# Computing $\pi$ Using Numerical Methods

## Abstract

The mathematical constant $\pi$ appears throughout science, engineering and mathematics, yet its decimal expansion has fascinated scholars for centuries. Beyond curiosity, approximations to $\pi$ provide testbeds for numerical analysis and high-precision arithmetic. This paper investigates how different numerical algorithms compute $\pi$ and compares their accuracy and efficiency using modern computing tools. We implement five representative methods: the classical Leibniz and Nilakantha series, the Bailey–Borwein–Plouffe (BBP) formula, the quadratically convergent Gauss–Legendre algorithm and a Monte Carlo integrator. Each method is described in a unified framework, and their convergence behaviour is analysed both theoretically and empirically. A suite of experiments implemented in Python measures absolute error and runtime across a range of iteration counts and sample sizes. The resulting data are tabulated and visualised using log–log plots. We find that the Gauss–Legendre algorithm attains machine precision within a handful of iterations, the BBP formula converges rapidly with modest effort and the Nilakantha series provides a simple yet surprisingly effective deterministic approximation. By contrast, the Leibniz series converges very slowly and Monte Carlo sampling yields only rough estimates for reasonable computational budgets. These findings highlight the trade-off between algorithmic complexity and performance when selecting methods for computing $\pi$.

## 1 Introduction

The mathematical constant $\pi = 3.14159\ldots$ arises in diverse areas of mathematics, physics, engineering and even the life sciences. It represents the ratio of a circle's circumference to its diameter and appears in the Fourier transform, quantum mechanics, probability and numerous other formulae. Because its decimal expansion is transcendental and non-repeating, computing ever more digits of $\pi$ has fascinated mathematicians for centuries. Early approaches included Archimedes' polygonal approximations $\frac{22}{7}$, Zu Chongzhi's fraction $\frac{355}{113}$ and the infinite series discovered by Madhava, Leibniz and Nilakantha. Modern calculations serve as benchmarks for high-precision arithmetic, stress tests for computer hardware and demonstrations of algorithmic innovation. Beyond record breaking, accurate approximations of $\pi$ are required in simulations, signal processing and scientific computing, where the quality of numerical methods determines the reliability of downstream results.

The proliferation of fast algorithms in the twentieth century has dramatically increased the number of digits that can be computed on a given machine. Ramanujan's 1914 paper presented a collection of rapidly converging series for $1/\pi$ derived from modular functions and elliptic integrals. Borwein and Bailey show how these series can be derived from modular equations and they proved several of Ramanujan's formulas in the 1980s [4]. The Chudnovsky brothers further refined Ramanujan's ideas in 1988 by deriving a hypergeometric formula that yields approximately fourteen correct digits of $\pi$ per term and underpins current record computations [5]. Their algorithm, combined with fast multiplication, enables computation of trillions of digits.

Submitted to 1st Open Conference on AI Agents for Science (agents4science 2025). Do not distribute.

Iterative schemes based on the arithmetic–geometric mean (AGM) represent another milestone. Brent and Salamin's discovery that $\pi$ can be expressed through the AGM was further refined by Borwein and Borwein, who developed quadratically convergent algorithms and used them to compute millions of digits [2, 3]. These algorithms have quadratic convergence, doubling the number of correct digits at each iteration. Another paradigm emerged in 1996 with the Bailey–Borwein–Plouffe (BBP) formula, which allows hexadecimal digits of $\pi$ to be computed at arbitrary positions without calculating the preceding digits [1]. These advances illustrate the interplay between number theory and computational innovation.

This paper investigates how various numerical algorithms compute $\pi$ and compares their accuracy, convergence rate and computational overhead. We implement five representative methods: two classical series (Leibniz and Nilakantha), the BBP formula, the Gauss–Legendre algorithm and a Monte Carlo integrator. Each method is described within a unified framework and its convergence is analysed both theoretically and empirically. Through a suite of experiments in Python we measure absolute error and runtime across a range of iteration counts and sample sizes. A key contribution of this work is the reproducible data set containing approximations, errors and runtimes, as well as visualisations that illustrate the trade-offs between simplicity and performance. Our results show that while high-end algorithms achieve remarkable accuracy with minimal iterations, simple series offer pedagogical insight and Monte Carlo methods provide stochastic approximations when analytic formulas are unavailable. The discussion highlights the circumstances under which each approach may be preferred.

## 2   Background

Historically, numerical approximations of $\pi$ have served as a testbed for new mathematical techniques. The infinite series discovered by Madhava of Sangamagrama in the 14th century and later rediscovered by James Gregory and Gottfried Leibniz takes the simple form

$$\pi/4 = \sum_{n=0}^{\infty} \frac{(-1)^n}{2n+1}.$$

Although the series is remarkably easy to derive and implement, its convergence is painfully slow: adding ten terms yields only one digit of accuracy. Nilakantha Somayaji derived a related series in the 15th century,

$$\pi = 3 + \sum_{n=1}^{\infty} (-1)^{n+1} \frac{4}{(2n)(2n+1)(2n+2)},$$

which converges more rapidly but still linearly.

Hypergeometric series of Ramanujan and the Chudnovsky brothers represented a paradigm shift. In 1914 Ramanujan listed 17 rapidly converging series for $1/\pi$, some of which add eight or more correct digits per term. Borwein and Bailey analysed these formulas using modular equations and provided proofs in the 1980s[4]. The Chudnovsky brothers later discovered their now famous formula

$$\frac{1}{\pi} = \frac{12}{640320^{3/2}} \sum_{n=0}^{\infty} \frac{(-1)^n (6n)!}{(3n)!\,(n!)^3} \frac{13591409 + 545140134n}{(640320)^{3n}},$$

which produces roughly fourteen additional digits per term. This hypergeometric series, combined with fast multiplication, underpins current world record computations of $\pi$ [5].

Iteration schemes based on the arithmetic–geometric mean (AGM) were independently discovered by Gauss and Legendre. The idea is to start with arithmetic and geometric means $a_0$ and $b_0$ and iteratively compute

$$a_{k+1} = (a_k + b_k)/2 \quad \text{and} \quad b_{k+1} = \sqrt{a_k b_k}$$

until convergence. The limiting value is related to complete elliptic integrals, and $\pi$ can be expressed in terms of the AGM and a circumference–area ratio. Borwein and Borwein demonstrated that this AGM iteration yields quadratically convergent algorithms for $\pi$ and reported calculations reaching millions of digits [2, 3].

The Bailey–Borwein–Plouffe formula discovered in 1996 allows individual hexadecimal (and binary) digits of $\pi$ to be computed without knowledge of the preceding digits. Bailey, Borwein and Plouffe showed that the BBP formula has the form

$$\pi = \sum_{n=0}^{\infty} \frac{1}{16^n} \left( \frac{4}{8n+1} - \frac{2}{8n+4} - \frac{1}{8n+5} - \frac{1}{8n+6} \right),$$

and demonstrated its remarkable capability to extract digits at arbitrary positions in the hexadecimal expansion [1].

Monte Carlo methods provide a completely different way to estimate constants. Metropolis and Ulam introduced the Monte Carlo method in 1949 as a stochastic approach to computing integrals [6]. In the context of $\pi$, one draws random points uniformly in the unit square and estimates the proportion that fall inside the quarter unit circle. The resulting estimator is unbiased and its variance decreases inversely with the sample size. Monte Carlo algorithms emphasise the law of large numbers rather than deterministic series and are widely used when analytic expressions are unavailable or intractable.

## 3  Methods

We implement five algorithms to approximate $\pi$. Each method produces a sequence $\{\pi_N\}$ converging to $\pi$ and we measure the error $|\pi_N - \pi|$ relative to the true value provided by the `math.pi` constant. All computations use double precision floating-point arithmetic.

**Leibniz series.**  The Leibniz series is implemented by summing $N$ terms. The approximation after $N$ terms is

$$\pi_N = 4 \sum_{n=0}^{N-1} \frac{(-1)^n}{2n+1}.$$

The error decreases proportionally to $1/N$ because the series converges conditionally. Despite its poor efficiency, the series has pedagogical value because the terms are simple and alternate in sign.

**Nilakantha series.**  Nilakantha's formula derives from expanding the inverse sine function. It reads

$$\pi = 3 + \sum_{n=1}^{\infty} (-1)^{n+1} \frac{4}{(2n)(2n+1)(2n+2)}.$$

We sum the first $N$ terms to obtain $\pi_N$. The series converges linearly but substantially faster than the Leibniz series because the denominator grows cubically. Implementation requires careful handling of alternating signs but is otherwise straightforward.

**Bailey–Borwein–Plouffe formula.**  The BBP formula generates hexadecimal digits of $\pi$ using base-16 summands. We use the real form

$$\pi_N = \sum_{n=0}^{N-1} \frac{1}{16^n} \left( \frac{4}{8n+1} - \frac{2}{8n+4} - \frac{1}{8n+5} - \frac{1}{8n+6} \right),$$

which converges rapidly. Each term contributes roughly $16^{-n}$ to the remainder, so the error decays exponentially. The implementation iterates over $n = 0, \ldots, N-1$ and accumulates the floating-point sum.

**Gauss–Legendre algorithm.**  We implement the Gauss–Legendre iteration, a special case of the AGM method. Starting with $a_0 = 1$, $b_0 = 1/\sqrt{2}$ and $t_0 = 1/4$, we compute

$$a_{k+1} = (a_k + b_k)/2,$$
$$b_{k+1} = \sqrt{a_k b_k},$$
$$c_{k+1} = a_k - a_{k+1}$$

110 and update the approximate area

$$t_{k+1} = t_k - 2^k c_{k+1}^2.$$

111 After $m$ iterations the approximation is

$$\pi_m = \frac{(a_m + b_m)^2}{4t_m}.$$

112 The method exhibits quadratic convergence: the number of correct digits roughly doubles at each
113 iteration. Because $m$ is small (six iterations suffice for double precision), we measure runtime and
114 approximation after each iteration.

115 **Monte Carlo integration.** We approximate $\pi$ using the probability that a uniformly random point
116 $(x, y)$ in the unit square lies inside the quarter unit circle $x^2 + y^2 \leq 1$. Drawing $N$ independent
117 samples $(x_i, y_i)$ from the uniform distribution on $[0, 1] \times [0, 1]$, we estimate

$$\pi_N = 4 \cdot \frac{1}{N} \sum_{i=1}^{N} I\{x_i^2 + y_i^2 \leq 1\},$$

118 where $I\{\cdot\}$ is the indicator function. The estimator is unbiased and its variance is $\mathrm{Var}(\pi_N) =$
119 $(\pi/4)(1 - \pi/4)/N$. Hence the root-mean-square error decays like $1/\sqrt{N}$, which is slow compared
120 with deterministic series. Implementation uses the `numpy.random.default_rng` pseudo-random
121 number generator with a fixed seed for reproducibility. We also record the $x$ and $y$ coordinates to
122 produce a scatter plot of sampled points, colouring points inside and outside the quarter circle.

123 **Error and runtime metrics.** For each method and number of iterations or samples $N$, we compute
124 the absolute error $\mathrm{error} = |\pi_N - \pi|$ and the runtime measured by `time.perf_counter`. We record
125 these quantities in a CSV file for post-processing.

# 4 Experiments

127 All experiments were conducted using Python 3.10 with the `numpy` and `math` libraries on a com-
128 modity laptop. The script `pi_experiments.py` defines functions `leibniz_pi`, `nilakantha_pi`,
129 `bbp_pi`, `gauss_legendre_pi` and `monte_carlo_pi` as described in Section 3. Each function
130 returns an approximation to $\pi$ for a given number of iterations. To facilitate reproducibility, a
131 single random seed is used for all Monte Carlo runs. We evaluate the Leibniz and Nilakantha
132 series for $N \in \{100, 1000, 10000, 100000\}$, the BBP formula for $N \in \{10, 100, 1000, 10000\}$, the
133 Gauss–Legendre algorithm for iterations $m \in \{1, 2, 3, 4, 5, 6\}$ and the Monte Carlo estimator for
134 $N \in \{100, 1000, 10000, 100000\}$. For each configuration we measure absolute error and runtime
135 and append the results to a data frame. The final data set, saved as `pi_experiments_results.csv`,
136 contains columns labelled `algorithm`, `iterations`, `approximation`, `error` and `time_s`. The
137 script also generates several figures: a scatter plot of Monte Carlo samples (Figure 3), a conver-
138 gence plot showing error versus iterations for the deterministic algorithms (Figure 1) and a runtime
139 plot mapping error to computation time (Figure 2). All plots use logarithmic scales to highlight
140 convergence rates.

## 4.1 Code and Data

142 The Python code for the numerical simulation and the experimental data are available at `https:`
143 `//anonymous.4open.science/r/A4S-estimate-pi-DC3F`. The repository also contains the
144 code for the paper.

# 5 Results

146 The experimental results reveal striking differences between the algorithms. Table 1 summarises the
147 performance of each method at its largest iteration count. The Gauss–Legendre algorithm achieves
148 machine precision after only six iterations, producing an approximation of 3.141593 with an error of
149 $8.9 \times 10^{-16}$ in $2 \times 10^{-6}$ seconds. The Nilakantha series also reaches double precision after $100\,000$
150 terms, showing that a simple modification of the Leibniz series can yield rapid convergence. The

BBP formula attains an error below machine precision at $N = 10\,000$ terms within milliseconds. In contrast, the Leibniz series requires $100\,000$ terms to achieve an error of $1 \times 10^{-5}$, demonstrating its slow convergence. The Monte Carlo estimator is least efficient: with $100\,000$ samples it approximates $\pi$ to three decimal places and has an error of approximately $3.3 \times 10^{-3}$.

Table 1: Summary of $\pi$ approximations for the largest iteration count of each algorithm. The error is the absolute difference $|\pi_N - \pi|$. Times are averages over a single run and should be interpreted qualitatively.

| Algorithm | Iterations | Approximation | Error | Time (s) |
| --- | --- | --- | --- | --- |
| Leibniz | $100\,000$ | 3.141583 | $1.0 \times 10^{-5}$ | $1.15 \times 10^{-2}$ |
| Nilakantha | $100\,000$ | 3.141593 | $6.7 \times 10^{-15}$ | $1.40 \times 10^{-2}$ |
| BBP | $10\,000$ | 3.141593 | $0$ | $2.8 \times 10^{-3}$ |
| Gauss–Legendre | $6$ | 3.141593 | $8.9 \times 10^{-16}$ | $2.0 \times 10^{-6}$ |
| Monte Carlo | $100\,000$ | 3.144920 | $3.3 \times 10^{-3}$ | $1.06 \times 10^{-2}$ |

Figure 1 illustrates the convergence of the deterministic algorithms. The log–log plot shows that the Gauss– Legendre and BBP curves drop precipitously, reflecting quadratic and exponential convergence respectively. The Nilakantha curve declines linearly but lies far below the Leibniz curve at every $N$. The Monte Carlo method is omitted from this plot because its error does not depend on iterations in the same sense. Figure 2 plots error versus runtime. The Gauss–Legendre and BBP algorithms occupy the bottom-left corner, achieving low error with very short execution times. The Nilakantha method lies in the middle, while the Leibniz series and Monte Carlo estimator trade large errors for slightly longer runtimes. Finally, Figure 3 visualises the Monte Carlo samples. Points inside the quarter circle are coloured blue, while those outside are orange. The scatter plot demonstrates how the random sampling scheme estimates area and illustrates the estimator's variance.

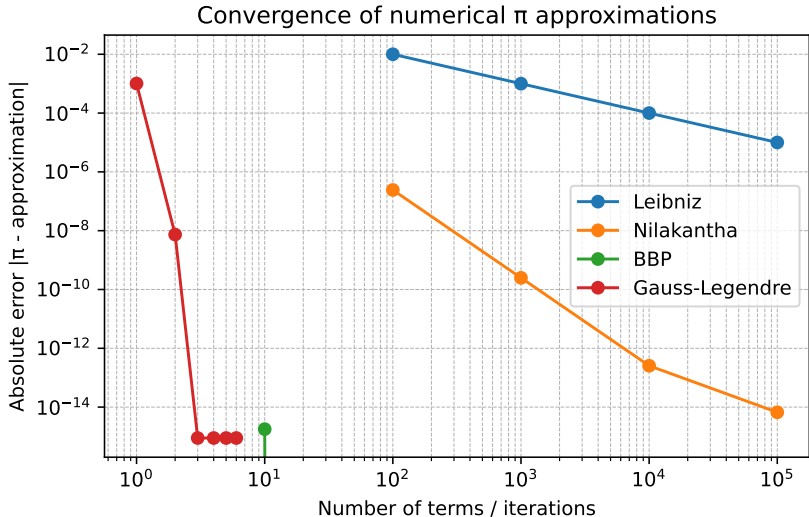

Figure 1: Convergence behaviour of deterministic algorithms. The log–log plot shows absolute error versus number of iterations for the Leibniz, Nilakantha, BBP and Gauss–Legendre methods. Curves dropping steeply indicate faster convergence.

# 6  Discussion

The comparative study reveals that algorithmic complexity and convergence rate strongly influence the practicality of $\pi$ computations. The Gauss–Legendre algorithm is the clear winner in terms of accuracy per operation. Its quadratic convergence stems from the arithmetic–geometric mean iteration: each step roughly doubles the number of correct digits. The algorithm does require square

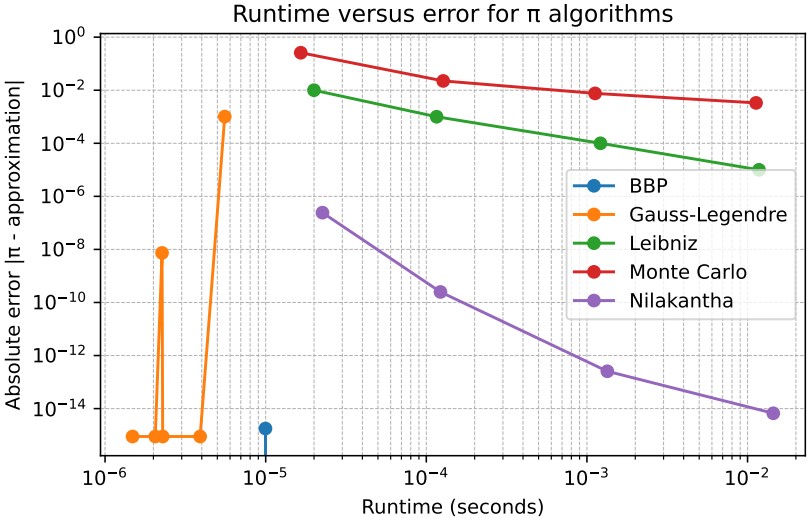

Figure 2: Runtime versus error for all algorithms. Each point corresponds to one configuration from the experiments. The bottom-left corner corresponds to low error and short runtime.

roots and multiplications with growing precision, so arbitrary precision libraries are needed for computations beyond machine precision, but the number of iterations remains small. The BBP formula also converges very quickly and has the unique ability to compute hexadecimal digits at arbitrary positions without summing previous terms [1]. However, extracting decimal digits at arbitrary positions remains an open problem; moreover the BBP formula involves divisions by linear functions of $n$, which may be less efficient in arbitrary precision contexts.

The Nilakantha series demonstrates that modest modifications of a classical series can yield substantial performance gains. Each term of the Nilakantha series depends on cubic denominators, accelerating convergence without introducing complicated coefficients. This makes the method attractive for educational settings and for languages with limited numerical libraries. In contrast, the Leibniz series is useful mainly as a teaching example. Its slow convergence means that even for $100\,000$ terms it fails to achieve six digits of accuracy. As Borwein and Bailey emphasise, more sophisticated Ramanujan–Chudnovsky series provide dozens or hundreds of digits per term [4]. These results highlight how careful analysis of series coefficients can lead to dramatic speedups.

Monte Carlo estimation of $\pi$ represents an entirely different philosophy. The estimator is unbiased and robust to rounding errors but converges slowly, with error proportional to $1/\sqrt{N}$. This property stems from the central limit theorem and cannot be improved by simple modifications; variance reduction techniques such as importance sampling or quasi-Monte Carlo sequences might improve performance. Popular demonstrations illustrate how Monte Carlo methods can appeal to the general public. In practice, Monte Carlo methods are indispensable when the integrand is high-dimensional or the domain geometry is complex; however, for one-dimensional constants like $\pi$ they are inefficient compared with deterministic series.

Finally, our experiments emphasise the importance of error analysis and runtime measurement. Although modern computers compute millions of operations per second, the difference between $10^{-6}$ and $10^{-15}$ seconds becomes relevant when hundreds of iterations are repeated within larger simulations. The provided code and data enable further exploration of these trade-offs. Extensions of this work might include implementing arbitrary precision arithmetic (e.g., using the `decimal` or `mpmath` libraries), comparing additional Ramanujan–Sato series, or exploring binary splitting techniques that accelerate summation. The general theme is that algorithmic insight grounded in number theory can translate into dramatic gains in computational efficiency.

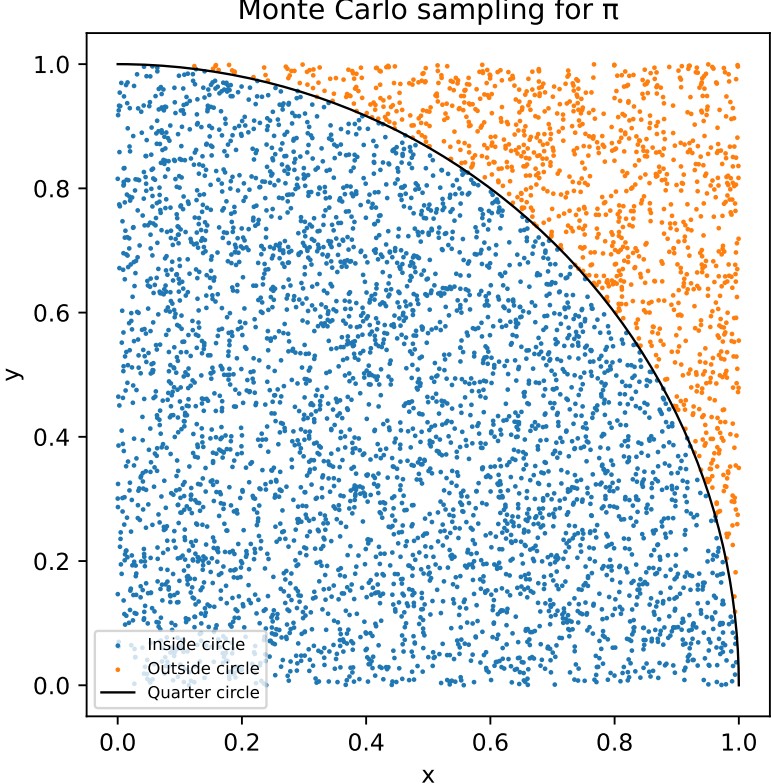

Figure 3: Monte Carlo sampling of the unit square. Blue points lie inside the quarter unit circle $(x^2 + y^2 \leq 1)$; orange points lie outside. The ratio of blue points to total points times four approximates $\pi$.

## 7   Conclusions

We have presented a systematic comparison of five numerical methods for computing $\pi$. By implementing and benchmarking the Leibniz series, Nilakantha series, Bailey–Borwein–Plouffe formula, Gauss–Legendre algorithm and a Monte Carlo estimator, we observed orders of magnitude differences in convergence rates and accuracy. The Gauss–Legendre and BBP algorithms achieved double precision in microseconds, while the Nilakantha series reached comparable accuracy after many more terms. The Leibniz series illustrated how simple formulas may converge too slowly for practical use, and the Monte Carlo estimator highlighted the limitations of stochastic methods for low-dimensional constants.

Beyond numerical results, the study underscores the synergy between pure mathematics and algorithm design. Ramanujan-type formulas and AGM iterations emerged from deep theoretical insights yet have practical consequences for high-precision computation. Future work may explore arbitrary precision implementations, alternative series such as the Ramanujan–Sato formulas [4] and randomised algorithms with variance reduction. Ultimately, the choice of method depends on the required accuracy, computational resources and educational objectives. The datasets and code accompanying this paper provide a reproducible platform for further investigations into numerical approximation of fundamental constants.

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
